# Perioperative Glial Fibrillary Acidic Protein Is Associated with Long-Term Neurodevelopment Outcome of Infants with Congenital Heart Disease

**DOI:** 10.3390/children8080655

**Published:** 2021-07-29

**Authors:** Michela Vergine, Luca Vedovelli, Manuela Simonato, Valentina Tonazzo, Alessio Correani, Elisa Cainelli, Dario Gregori, Massimo A. Padalino, Paola Cogo

**Affiliations:** 1Division of Pediatrics, Department of Medicine, University Hospital S Maria della Misericordia, University of Udine, Piazzale Santa Maria della Misericordia, 15, 33100 Udine, Italy; michela.vergine@asufc.sanita.fvg.it; 2Unit of Biostatistics, Epidemiology and Public Health, Department of Cardiac, Thoracic, Vascular Sciences and Public Health, University of Padova, Via Loredan, 18, 35131 Padova, Italy; luca.vedovelli@ubep.unipd.it (L.V.); dario.gregori@ubep.unipd.it (D.G.); 3PCare Laboratory, Fondazione Istituto di Ricerca Pediatrica, “Citta’ della Speranza”, Corso Stati Uniti, 4F, 35127 Padova, Italy; 4Department of Women’s and Children’s Health, University of Padova, Via Giustiniani, 3, 35128 Padova, Italy; tonazzo.v@gmail.com; 5Division of Neonatology, Polytechnic University of Marche and “G. Salesi” Children’s Hospital, Via Corridoni, 11, 60123 Ancona, Italy; alessio.correani@gmail.com; 6Department of General Psychology, University of Padova, Via Venezia, 8, 35131 Padova, Italy; cainelli.elisa@gmail.com; 7Pediatric and Congenital Cardiac Surgery, Department of Cardiac-Thoracic-Vascular Sciences and Publich Health, University of Padova, Via Giustiniani, 2-35128 Padova, Italy; massimo.padalino@unipd.it; 8Department of Medicine, University Hospital S Maria della Misericordia, University of Udine, Piazzale Santa Maria della Misericordia, 15, 33100 Udine, Italy; paola.cogo@uniud.it

**Keywords:** congenital heart disease, cardiopulmonary bypass, children, neuropsychological outcome, glial fibrillary acidic protein

## Abstract

Background: Brain injury, impaired brain maturation, and long-term neurodevelopmental disorders are common in infants with congenital heart diseases (CHD). We aimed to assess whether plasma glial fibrillary acidic protein (GFAP) can predict neurodevelopmental anomalies in CHD infants operated on cardiopulmonary bypass (CPB). Methods: We measured plasma GFAP in 38 infants at multiple CPB phases. Cognitive, neuropsychological, and psychopathological functioning were assessed 5.7 ± 2.2 years after surgery. We identified an impaired global neurodevelopmental index (NDI) when at least two domains were abnormal. The relationships between NDI, GFAP, and clinical variables were explored with non-supervised feature selection methods and modeled with a nested non-linear logistic regression. Results: Intelligence quotient scores were within the normal range in 84% of children, whereas 58% showed an abnormal NDI, with the greatest impairments in the psychopathological area. The plasma GFAP peak was 0.95 (0.44–1.57) ng/mL, and it was correlated with age, weight, duration of surgery phases, and CPB minimum temperature. In the regression model, the GFAP peak was associated with an impaired NDI with a possible flexible point toward NDI impairment at 0.49 ng/mL, keeping constant ICU stay, CPB duration, CHD anatomy, weight, and CPB minimum temperature. Conclusion: GFAP is a promising early marker of abnormal long-term neuropsychological development.

## 1. Introduction

Congenital heart disease (CHD) is the most frequent congenital malformation with an estimated prevalence of 8 cases out of 1000 [1]. As survival has greatly improved over the past few decades, medical attention has been focused on long-term outcomes of patients with CHD [2]. More than 50% of children with congenital heart disease (CHD) are reported to present with neurodevelopmental impairments later in life, mainly associated with abnormal cognition, language and communication, motor and executive functioning, behavior, and social interaction domains [3]. These impairments are associated with a global reduction in quality of life of children with CHD and their families [2,4]. The underlying cause is multifactorial and related to fetal, perinatal, and perioperative factors, as it is well demonstrated by several neuroimaging studies performed in the neonatal era, which show the presence of congenital and acquired brain injury, occurring both pre- and postoperatively [5,6,7]. Preoperative brain insults are mainly due to genetic abnormalities, brain dysmaturation, and anomalies of cerebral blood flow during pregnancy, and thus they are not preventable [3,6]. Conversely, brain insults occurring during and post-cardiac surgery could be preventable by optimizing the neuroprotective strategies applied perioperatively.

A bright marker of brain injury is represented by glial fibrillary acidic protein (GFAP), the main protein of mature astrocytes, which is promptly released into the bloodstream after an acute neurologic insult [8]. An increasing number of studies analyzed the GFAP blood values during and after surgery in children with CHD [9,10,11], but only a few have investigated its potential role in predicting adverse neurodevelopmental outcomes [12,13,14]. We have previously published that neurologic adaptive functioning, assessed by the Vineland Adaptive Behavior Scale, is associated with increased plasma levels of GFAP measured during cardiac surgery in infants with CHD [13]. However, such evaluation is based on a questionnaire completed by the parents with no direct measure of the children’s skills. Therefore, we performed a comprehensive cognitive, neuropsychological, and psychopathological assessment after 5.7 ± 2.2 years from surgery in a cohort of CHD children operated on cardiopulmonary bypass (CPB) to assess the role of GFAP in predicting long term neurodevelopmental outcome. 

## 2. Materials and Methods

This was a prospective, observational, single-center study in children with CHD undergoing elective cardiac surgery from 2010 to 2017. Inclusion criteria and the study design have previously been reported in detail [10,11,13]. Briefly, we studied children on preoperative stable hemodynamic condition who required cardiac surgery with CPB time >60 min on hypothermia and aortic cross-clamp time >20 min, with written informed consent. Children with preoperative signs of organ failure and with chromosomal, genetic, or neurological abnormalities were excluded.

### 2.1. Perioperative Management

After the induction of anesthesia (fentanyl 5 μg/kg, thiopental 3 mg/kg or midazolam 0.2 mg/kg, and vecuronium bromide 0.1 mg/kg), infants were intubated, and a central venous catheter was placed. General anesthesia was obtained with fentanyl 50 μg/kg, cisatracurium besylate 3 mg/kg, and midazolam 3 mg/kg, infused at 1 mL/h or 2 mL/h, based on patient body weight (less or greater than 5 kg, respectively). After heparin 300 U/kg administration (activated clotting time (ACT) target = 480 s), arterial and venous cannulation were performed, and CPB was initiated. Hematic prime was used to maintain hematocrit between 25% and 30%. Temperature was monitored with nasopharyngeal and rectal probes. Hypothermia during CPB was defined as mild (35–30.1 °C), moderate (30–25.1 °C), and deep (25–15.1 °C). Whenever required, hematic cardioplegia, aortic cross-clamp, and DHCA or selective regional cerebral perfusion were used; adequate CPB flows were calculated based on body surface area, cardiac index, and minimal body temperature reached. 

During surgery, cerebral regional oxygen saturation (rO_2_) was recorded every minute by near-infrared spectroscopy, and blood gas analysis was measured every 20 min. At the end of the surgery, children were rewarmed to 36 °C. 

We prospectively collected detailed demographic, clinical, and surgical data during hospital admission. Before discharge, all children underwent a neurological assessment to rule out any acquired neurological injury.

Children were classified according to the underlying CHD anatomy (Clancy classification) [15] and physiopathology (Rigby classification) [16] and also according to the STAT category [17], which defines the complexity and mortality risk of the surgical procedure. 

Between 2018 and 2019, all children aged 4 to 8 years were offered a comprehensive cognitive, neuropsychological, and psychopathological evaluation. We excluded children living far from our hospital (>150 km) since the neuropsychological evaluation required at least two separate appointments at our institution; those who did not speak Italian, as all neurodevelopmental tests were validated in Italian; and children with newly acquired neurological comorbidities. The study was approved by the local Institutional Review Board and Ethics Committee (Protocol number 3142/AO/14; Padova Hospital, Padova, Italy).

### 2.2. Sample Collection and GFAP Analysis

Blood samples were collected at anesthesia induction, CPB start, end of hypothermia, end of rewarming, and end of CPB before modified ultrafiltration. They were spun at 1400× *g* for 10 min, and plasma was aliquoted and stored at −80 °C; GFAP concentration was determined by ELISA kit (BioVendor, Brno, Czech Republic). Non-detectable GFAP values were recorded as half of the limit of detection (0.02 ng/mL). 

### 2.3. Neurodevelopmental Testing

Cognitive and neuropsychological functioning was assessed by a child psychologist. The test battery required nearly 3 h, divided into two meetings on successive days to be completed; several breaks were planned within each evaluation. A quiet environment was maintained throughout the assessments. Cognitive, neuropsychological, and psychopathological functioning was assessed by tests that were selected based on developmental challenges that are common in children with CHD. 

All measures were validated and have normative values based on a healthy Italian population. We used the Wechsler Preschool and Primary Scale of Intelligence III [18] test or the Wechsler Intelligence Scale for Children IV [19,20] to evaluate general cognitive performance. Results on these two tests include a general measure of intelligence quotient (IQ) with quotients for verbal and nonverbal performance and processing speed. The following cognitive domains were assessed during the neuropsychological testing: language, using the naming test, idiomatic sentences, emotional prosody [21]; attention, using the visual and auditory attention tests of the NEPSY-II [22]; executive functions, using the graphic fluency of the NEPSY-II [22]; social skills, using the theory of mind and the emotional recognition tests of the NEPSY-II [22]. Each child’s psychopathological profile was explored by performing an in-depth psychological interview with the parents. Anamnestic and academic information and psychomotor development were investigated. Parents also completed two questionnaires. The Child Behavior Checklist (CBCL) is a multiaxial empirically-based set of measures that includes parent-, self-, and teacher-report versions for assessing social competence and emotional/behavioral problems in children [23]. The parent report comprises items assessing the child’s emotional, behavioral, and social problems over the past six months. The items produce eight original empirically derived syndrome scales for ages 6–18: social withdrawal, somatic complaints, anxiety/depression, social problems, thought problems, attention problems, rule-breaking behavior, and aggressive behavior. The items produce seven scales for ages 1 ½ −5: social withdrawal, somatic complaints, anxiety/depression, emotional reactive, sleep problems, attention problems, and aggressive behavior. Both produce two broadband scales: externalizing problems and internalizing problems. Conner’s Rating Scales-Revised (CRS-R) and Conner’s Parent Rating Scales—Long Version (CPRS-R:L) report parent ratings of child behaviors involving problems in seven psychopathological areas: oppositional, inattention, hyperactive, anxious–shy, perfectionism, social problems, and psycho-somatic [24].

### 2.4. Statistical Analysis

Results are expressed as means and standard deviations (SD) and medians (interquartile ranges). 

Scores of the cognitive, neuropsychological, and psychopathological instruments were age-corrected and converted into z scores and scaled scores (neuropsychological tests), quotients (cognitive tests), or T scores (psychopathological questionnaires), as appropriate, using published normative data. 

Results of the WPPSI-III and of the WISC-IV were converted in quotients with a mean of 100 and SD of 15. Impairment was defined as a quotient lying 2 SD below the mean (<70), the borderline was defined between 70 and 85. The z scores indicate the deviation from the mean population score, which was set to 0, SD 1. A z score of −2 (or less) comprised 2.5% of the normal distribution and is considered to be significantly lower than average. Scaled scores indicate the deviation from the mean population score, which was set to 10, SD 3. A standard score of 5 (or less) is considered to be significantly lower than average. T scores indicate the deviation from the mean population score, which was set to 50, SD 10. A T score of 65 (or more) indicates a clinically relevant condition.

The cognitive, neuropsychological, and psychopathological results were summarized in a global index. A neurodevelopmental index (NDI) was calculated from the impairments of single functions. A child had an altered NDI if he/she obtained at least two severe impairments at neuropsychological tasks (<2 SD), IQ < 85, and/or a clinically relevant score on psychopathology questionnaires (>65 T scores).

Linear correlations were assessed by Spearman’s Rho, and *p*-values were adjusted for multiple comparisons with the Holm method. Feature selection (i.e., which were the most important variables for the classification NDI normal or altered) was implemented with a random forest-based method (Boruta). Furthermore, an exhaustive search for the best subsets of the variables to include in the logistic regression model was implemented. Variables were then selected based on the exploratory analysis results and on clinical relevance to building a logistic non-linear regression model with a single cubic spline on the GFAP max variable. The position of the spline was determined iteratively with the leave-out-one cross-validation of each model obtained using different splines. The spline that gave the model with minimum cross-validation error was then selected to be included in the final model. Finally, we constructed two nested logistic regression models, one including GFAP max as a linear predictor and with GFAP max non-linearly modeled with a cubic spline, to evaluate if GFAP was associated with NDI. Data were analyzed and graphed using R software v. 4.0. Code used for the analysis and a list of r-packages used are available as Appendix A.

## 3. Results

### 3.1. Anatomical, Clinical, and Surgical Characteristics

We enrolled 38 children from May 2018 to December 2019. Twenty-one out of 38 patients were part of a previous publication [13], with the Vineland Adaptive Behavior Scale applied 18 months after surgery. 

All CHD children were inborn with adequate growth for gestational age and with uneventful perinatal history. Table 1 shows the study’s clinical and surgical data. The most frequent CHD was a transposition of the great arteries (*n* = 7) and single ventricle physiology (*n* = 7), followed by tetralogy of Fallot (*n* = 6), hypoplastic left heart syndrome (*n* = 5), ventricular septal defect (*n* = 5), and partial atrioventricular septal defect (*n* = 4). The remaining four CHD were one cor triatriatum, one interrupted aortic arch, one truncus arteriosus, and one left ventricle-to-aorta tunnel. Twenty-four children (63%) underwent a single cardiac surgery on CPB at the time of neurodevelopmental follow-up. Thirty-three children were studied at their first cardiac surgical intervention (87%), three at the second, and two at the third cardiac surgery. 

No one had signs of a low cardiac output state postoperatively. Twelve children (32%) had postoperative complications, and two needed a reintervention. All children were discharged after a median hospital stay of 11 days (IQR 7–14).

According to Clancy classification [17], 25 children (*n* = 66%) had 2 ventricles and no aortic arch obstruction (Class 1), 2 (5%) were CHD with 2 ventricles and aortic arch obstruction (Class 2), 4 (11%) were single ventricle with no aortic arch obstruction (Class 3), and 7 (18%) were single ventricle with aortic arch obstruction (Class 4).

According to Rigby et al. [16], 19 children (50%) had increased pulmonary blood flow (Class 1), 8 (21%) had reduced pulmonary blood flow (Class 2), and 11 (29%) had increased pulmonary venous pressure (Class 3).

### 3.2. GFAP and Neurodevelopmental Outcome

Before surgery, GFAP was undetectable (<0.02 ng/mL) in 30 children (79%). Of the remaining eight patients, four had values below 0.10 ng/mL and four (all with a single-ventricle anatomy) between 0.10 and 0.20 ng/mL. 

The mean peak GFAP value was 1.42 ± 1.69 ng/mL, with a median of 0.95 ng/mL (IQR 0.44–1.57). Details of plasma GFAP levels during surgery are reported in Figure 1. GFAP peaked at the end of hypothermia in 8 children (21%), at the end of the rewarming in 11 (29%), and at the end of CPB in 19 (50%).

The cognitive, neuropsychological, and psychopathological assessment was performed at a mean age of 6.5 ± 2.0 years, after 5.7 ± 2.2 years from surgery: 12 children (31%) were in the pre-school age range. Twenty-two children (58%) resulted in having an abnormal NDI, with the greatest impairment in the psychopathological area (Table 2).

### 3.3. Correlation and Regression Analysis

There were significant negative correlations of the GFAP max value with weight, age, and minimum temperature.

The duration of surgery, CPB, hypothermia, and rewarming were positively correlated with the GFAP max value (Figure 2).

Random-forest-based feature selection classified the GFAP max, minimum temperature, and weight as the most important variables for the classification of the patients with normal and impaired NDI (Figure 3).

A regression subsets exhaustive search confirmed minimum CPB temperature and GFAP max as important variables to determine the NDI status, along with days spent in ICU and Rigby classification

We selected GFAP max, ICU days, CPB duration, Clancy classification (for the best performance of the final model in respect to the Rigby classification), weight, and minimum CPB temperature as predictors to include in the logistic regression model with NDI normal or impaired as the outcome. 

While fitting the final logistic regression model, we found that the relationship between GFAP and NDI was non-linear, with a tendency for all patients with a higher GFAP max to have an impaired NDI. We thus decided to allow GFAP max the possibility to vary non-linearly. We fitted the above model varying the position (from 0.1 to 2) of a single cubic spline for GFAP max, and we cross-validated each model. The model with the minimum cross-validation error was the one with the spline positioned at GFAP max = 0.49 ng/mL.

We finally fitted two nested models, one with GFAP max as a linear predictor and one as GFAP non-linearly modeled with a cubic spline at 0.49 ng/mL. The two models were then compared (Chi-squared test), and GFAP max resulted significantly (*p* = 0.02) associated to NDI, keeping constant the other predictors (ICU days, CPB duration, Clancy classification, weight, and minimum CPB temperature). The area under the curve (AUC) of the receiver operating characteristic (ROC) curve was 0.85 (95% CI, 0.71–0.99) for the final model.

The peak GFAP value was >0.49 ng/mL in 55% of children belonging to STAT 1 and 2 categories, and in 89% of children belonging to STAT 3 to 5, whereas there was no difference in the percentage of abnormal NDI (73 vs. 75% respectively). Two of the 20 children belonging to STAT 1 and 2 showed abnormal NDI without peak GFAP >0.49 ng/mL.

The percentage of children with a peak GFAP value >0.49 ng/mL and with an abnormal NDI did not differ among the three hemodynamic groups, according to Rigby et al. [16].

According to Clancy classification [15], 80% of children with two ventricles and no aortic obstruction (Class 1, *n* = 25) had a peak GFAP value >0.49 ng/mL and an abnormal NDI; 100% of the children belonging to Clancy Class 2 (two ventricles and aortic obstruction, *n* = 2) had the same outcome, whereas none in Clancy Class 3 (one ventricle no aortic obstruction, *n* = 4) and 80% in Clancy Class 4 (one ventricle and aortic obstruction *n* = 7) (Appendix A) did. 

## 4. Discussion

In this study, we showed that the plasmatic level of GFAP measured during cardiac surgery is associated with an abnormal neurodevelopmental assessment at 4–8 years of age, even in the presence of a normal IQ. 

GFAP has been recently recognized by the US Food and Drug Administration as a blood test biomarker for brain injury and acute neurological damage [25]. We have previously demonstrated that during cardiac surgery, GFAP is not detectable at anesthesia induction in most CHD children, but its plasma level rises up soon after starting CPB [10,13]. Moreover, in these CHD children, GFAP was associated with impaired communication skills measured with the Vineland Adaptive Behavior Scale [13], as also reported by Graham et al. [12]. Increased GFAP plasma levels have been associated with neurological disability, defined as intracranial hemorrhage, brain infarction, cerebral edema, or brain death, in children receiving extracorporeal membrane oxygenation [14] and with abnormal functional outcomes in neonates with birth asphyxia and born preterm [26]. Serum GFAP levels during the first week of life were increased in neonates with hypoxic-ischemic encefalopathy and were predictive of brain injury on MRI [27].

Higher-order cognitive function requires proper connectivity and communication between neurons locally and distally throughout the brain. WM tracts represent the structural foundation of brain connectivity and their primary constituent is represented by myelinated axons. WM development is a complex and long process, starting in the first months of gestation, involving oligodendrocytes maturation and development. Diffused WM injuries are a key component of pediatric neurodevelopmental disorders, including neuropsychological and behavioral disorders. GFAP is part of the astrocyte cytoskeleton; thus, it is specific to the central nervous system, and its expression is higher in white matter compared to grey matter astrocytes [28]. The inflammatory cascade, initiated by different mechanisms of brain injury occurring during cardiac surgery with cardiopulmonary bypass use, may lead to the disruption of the blood–brain barrier and the release of GFAP into the bloodstream.

Since we could not demonstrate an association between GFAP increase and overt brain damage, we speculated that GFAP plasma levels could reflect an acute and subtle white matter injury induced by reactive oxygen species during CPB and sudden change in brain temperature [29]. In fact, we found a mean peak GFAP level of 1.46 ng/mL, recorded mainly at the end of the hypothermia-rewarming phase or at the end of CPB run, as previously shown [10,11,13], while it was mostly undetectable at anesthesia induction, suggesting a prominent GFAP release during the acute changes of brain temperature. The same findings were reported by other studies in which moderate or deep hypothermia, but also the rewarming phase, were associated with increased GFAP, neuronal cytotoxicity, endothelial dysfunction, inflammation [11,30], and loss of vascular brain autoregulation [31]. 

The prevalence and severity of developmental disabilities increase with the complexity of CHD [32]. Multiple factors not related to surgery, such as degree of cyanosis, genetic syndromes, brain malformations, comorbidities, brain maturity, perinatal or subsequent hypoxemic–ischemic events, prenatal brain injury, and socioeconomic and environmental factors, contribute to the neurodevelopmental outcome of CHD children [3,7]. To this end, we accurately selected infants with no documented preoperative hypoxic-ischemic events who underwent elective cardiac surgery with no signs of low cardiac output syndrome. By doing this, we aimed to study the most critical intraoperative events that could contribute to an altered neurodevelopmental outcome. 

The peak GFAP plasma level, especially if greater than 0.49 ng/mL, resulted as a significant risk factor for abnormal neurodevelopment at long-term follow-up especially in the social and behavioral area [32], keeping constant the other predictors (ICU days, CPB duration, Clancy classification, weight, and minimum CPB temperature). Interestingly, 0.49 ng/mL is very close to the previously reported threshold value of 0.46 ng/mL, which was able to discriminate children with brain injuries during extracorporeal membrane oxygenation (ECMO) [33]. 

After moderate hypothermia, rewarming to 36 °C was associated with a significant increase in the average cerebrovascular pressure reactivity index, indicating temperature-dependent hyperemic derangement of cerebrovascular reactivity [34]. Increased neuronal apoptosis has been demonstrated during rewarming from hypothermia in a swine model of hypoxic-ischemic cardiac arrest. Neuroapoptosis in the piriform cortex was also worse in hypoxic-ischemic, rewarmed piglets than in naive/sham piglets. The rate of rewarming was positively correlated with worse neuronal injury [35].

This study had some limitations. First, the small sample size may have hidden some significant results. In addition, the cohort was quite heterogeneous regarding both age at follow-up and CHD diagnoses with different severity and associated mortality and morbidity risk.

Strengths of this study include the comprehensive neurologic and neuropsychological assessment that provided a complete profile of neurodevelopment. This follow-up study included individual in-person evaluations and outcome measures not based only on parent-reporting, which can be biased and may overestimate a child’s capabilities [36]. Moreover, enrollment from a single center helped to eliminate confounding factors regarding anesthesiologic and surgical management.

## 5. Conclusions

We demonstrated that GFAP during CPB could be a reliable marker to predict altered neurodevelopmental outcomes in CHD children. CPB phases characterized by body temperature changes seem to be the most dangerous period for developing neurological insult, traced as GFAP release. Further data are needed to develop neuroprotective strategies during CPB in CHD children.

## Figures and Tables

**Figure 1 children-08-00655-f001:**
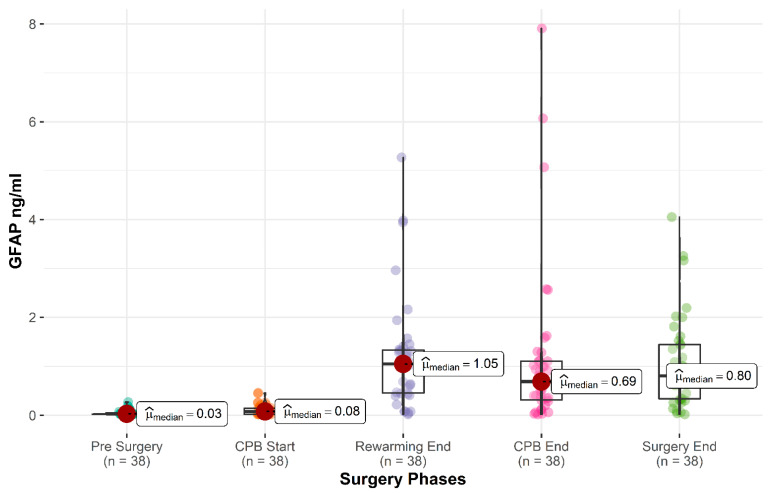
GFAP plasma level at prespecified intraoperative time points. Boxviolin plots for GFAP plasma level. Boxes represent the interquartile range. CPB, cardiopulmonary bypass; GFAP, glial fibrillary acidic protein.

**Figure 2 children-08-00655-f002:**
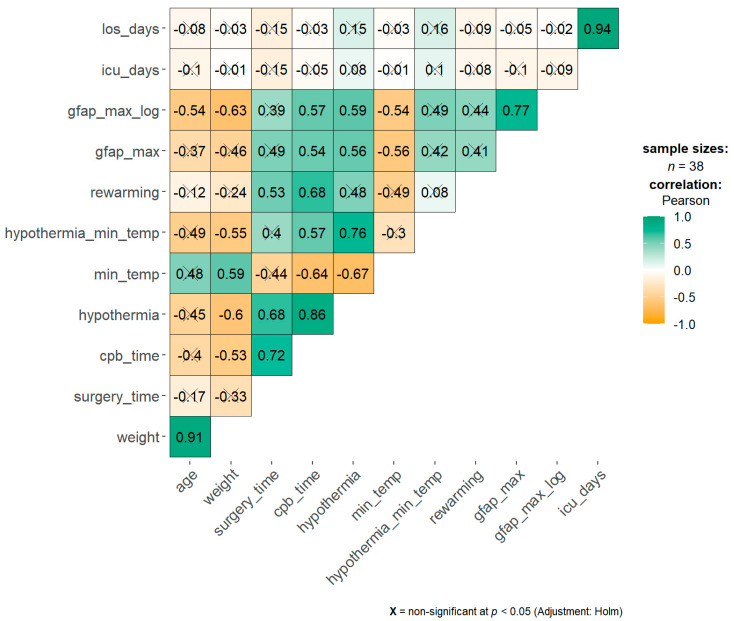
Correlation matrix among surgery parameters and GFAP max value. Spearman’s Rho value is reported inside each square. Color intensity is related to the Rho coefficient: orange as negative correlation, green as positive. Crossed Rho values are non-significant. *p*-values were adjusted for multiple comparisons. *los*: length of stay, *icu*: intensive care unit, *cpb*: cardiopulmonary bypass.

**Figure 3 children-08-00655-f003:**
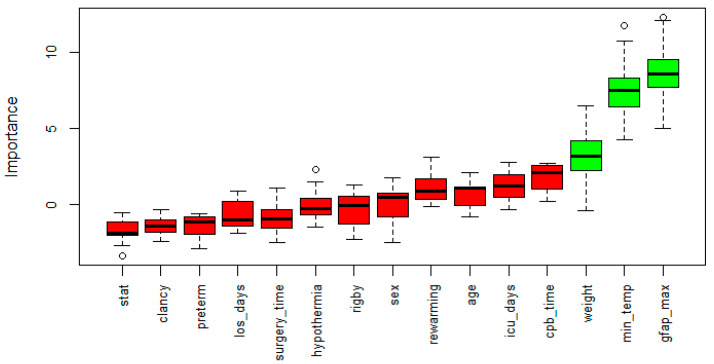
Variable importance for impaired and normal NDI classification. A random-forest-based feature selection method (Boruta) was implemented to obtain the most important variables able to classify patients for their NDI. Green boxes depict the obtained three significant (*p* < 0.01) variables. *los*: length of stay, *icu*: intensive care unit, *cpb*: cardiopulmonary bypass. Dots represents outlier values defined as above the third quartile +1.5x interquartile range.

**Table 1 children-08-00655-t001:** Clinical and surgical data.

Characteristic	Number	Mean ± SD	Median (IQR)
Sex (F/M)	16/22		
Gestational Age (weeks)		37.9 ± 3.2	38.0 (38.0–40.0)
Term/Preterm (GA < 37 weeks)	32/6		
Neonatal weight (grams)		3058 ± 648	3228 (2798–3468)
Head circumference (cm)		33.7 ± 1.7	34.0 (34.0–34.5)
Percentile of HC (%)		47 ± 25	43 (34–61)
Apgar 1′		8.4 ± 0.7	9.0 (8.0–9.0)
Apgar 5′		9.2 ± 0.7	9.0 (9.0–10.0)
SpO2 (%)		89.2 ± 8.6	88.5 (85.0–98.0)
STAT category:			
1	5 (13.2%)		
2	15 (39.5%)		
3	7 (18.4%)		
4	8 (21%)		
5	3 (7.9%)		
Age at surgery (months)		14.9 ± 20.2	4.0 (0.0–23.8)
<1 month of age (*n*, %)	12 (31.6%)		
1–12 months of age (*n*, %)	14 (36.8%)		
>12 months of age (*n*, %)	12 (31.6%)		
Weight at surgery (kg)		7.6 ± 4.9	5.8 (3.5–11.9)
Time (minutes)			
Surgery		249 ± 71	248 (196–281)
CPB		123 ± 49	120 (87–152)
Aortic clamp (*n* = 30)		62 ± 26	63 (41–79)
Circulatory arrest (*n* = 4)		26 ± 27	24 (3–46)
Duration of Hypothermia		75 ± 34	70 (50–90)
Duration of Rewarming		33 ± 21	30 (20–40)
Minimal body temperature (°C)		30.2 ± 3.9	32.0 (28.0–32.7)
rO_2_ during CPB (%)		55.9 ± 10.4	56 (48–60)
rO_2_ <45% during CPB (%)		19.6 ± 27.5	10 (0–20)
Intubation (hours)		13 ± 267	50 (26–90)
Admission time (days)	Data are given as mean ± SD, median (IQR), or *n* (%). CPB, cardiopulmonary bypass; ICU, intensive care unit; NA, not available.		
ICU		7 ± 13	4 (2–5)
Total		15 ± 18	12 (7–14)

Data are given as mean ± SD, median (IQR), or *n* (%). CPB, cardiopulmonary bypass; ICU, intensive care unit.

**Table 2 children-08-00655-t002:** Neuropsychological testing results.

Neuropsychological Assessment	
NDI abnormal: *n* (%)	22 (57.9%)
Total IQ	99.7 ± 16.7
*Pathological cases*: (*n*, %)	
IQ	6 (15.8%)
Language	1 (2.6%)
Attention	14 (36.8%)
Executive function	2 (5.3%)
Social abilities	13 (34.2%)
Psychopathology	17 (44.7%)

Data are given as mean ± SD or *n* (%). IQ, intelligence quotient; NDI, neurodevelopment index.

## Data Availability

On request.

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
