# Peer review of "Perioperative Glial Fibrillary Acidic Protein Is Associated with Long-Term Neurodevelopment Outcome of Infants with Congenital Heart Disease"

_children, 2021, doi:10.3390/children8080655_

Round 1

Reviewer 1 Report

The authors describe the relationship of GFAP to various behavioral and cognitive dysfunctions using a very novel methodology. Monitoring GFAP levels during cardiopulmonary bypass in cardiac surgeries has the potential to identify serum level cut points at which time neurodegenerative pharmacologic countermeasures may be administered sequentially in order to prevent long-term deficits.

Since it is known that various medications used during anesthesia can influence the extent of neurodegeneration, the manuscript could be strengthened by an elucidation of what agents patients received, such as ketamine, benzodiazepines, dexmedetomidine, nitric oxide, and barbiturates.

The manuscript is well-written and very interesting. The title and abstract are descriptive. Key words are appropriate. References are in mdpi style.

Well done! Thank you for the opportunity to review your fine contribution to the literature.

Author Response

Comment: The authors describe the relationship of GFAP to various behavioral and cognitive dysfunctions using a very novel methodology. Monitoring GFAP levels during cardiopulmonary bypass in cardiac surgeries has the potential to identify serum level cut points at which time neurodegenerative pharmacologic countermeasures may be administered sequentially in order to prevent long-term deficits.

Since it is known that various medications used during anesthesia can influence the extent of neurodegeneration, the manuscript could be strengthened by an elucidation of what agents patients received, such as ketamine, benzodiazepines, dexmedetomidine, nitric oxide, and barbiturates.

Reply: We thank the reviewer for the suggestion, we revised the methods section and we add the “perioperative management” paragraph (2.1. lines 90-108).

Comment: The manuscript is well-written and very interesting. The title and abstract are descriptive. Key words are appropriate. References are in mdpi style. Well done! Tank you for the opportunity to review your fine contribution to the literature.

Reply: we thank te reviewer.

Reviewer 2 Report

Dear Authors,

I would like to thank authors for interesting presentation of their work on GFAP a marker expressed primarily by astrocytes in the CNS that may influence long-term brain development in pediatric cardiac surgery. I find the manuscript well written and interesting but the low sample of study cause the far-reaching conclusions presented by authors. 

I would like to admit, that the manuscript is well written. The study is of a very low sample which may not, in my humble opinion, allow for far-reaching conclusions. Its more preliminary report that study presentation.

Moreover, I have some more questions:

  1. Did any analysis of risk for GFAP genetic mutations that could interfere with its levels were taken into consideration?
  2. Reinterventions in two cases could probably influence with GFAP by deficiency brain supply by hypoperfusion as I suspect the main indication was excessive bleeding. May I ask authors for any comment?
  3. During the surgery, what was the minimal hematocrit? Was any relation found between risk for GFAP and hemodilution?
  4. During the surgery, were there any electric brain activity (EEG waves) monitoring performed like CSA?

Kind Regards

Author Response

Comment: Dear Authors, I would like to thank authors for interesting presentation of their work on GFAP a marker expressed primarily by astrocytes in the CNS that may influence long-term brain development in pediatric cardiac surgery. I find the manuscript well written and interesting but the low sample of study cause the far-reaching conclusions presented by authors. I would like to admit, that the manuscript is well written. The study is of a very low sample which may not, in my humble opinion, allow for far-reaching conclusions. Its more preliminary report that study presentation.

 Reply: We thank the reviewer for the kind comments. We are sorry that the reviewer did not find out manuscript suitable for publication. We are aware that the sample size is limited, but we think that our paper is adding some important knowledge in the field, because of the rigorous and meticolous design of the study, and mostly of the innovative message which aims to  focus on quality of life of children with CHD, rather than only on survival, as it has been done for decades.Also, the presented results are in line with the current literature, and add more information to current knowledge. Last, our statistical model was able to obtain the same threshold for GFAP obtained in la larger cohort and in a different disease. This is of paramount importance since it suggests that this kind of threshold could be generalized to other diseases.

Comment: Moreover, I have some more questions: Did any analysis of risk for GFAP genetic mutations that could interfere with its levels were taken into consideration?

Reply: We thank the reviewer for the comment. We are aware that mutations in GFAP gene leads to Alexander Disease which is characterized by progressive accumulation in astrocytes of GFAP aggregates (Ciammola A doi: 10.3389/fneur.2019.01124). Alexander disease is extremely rare with a prevalence of 1 in 2.7 million and even if we did not screen our patient for this neurological disorder, we think that the risk for GFAP genetic mutations in our study cohort is extremely low. We can take it into consideration for the next studies.

Comment: reinterventions in two cases could probably influence with GFAP by deficiency brain supply by hypoperfusion as I suspect the main indication was excessive bleeding. May I ask authors for any comment? 

Reply: We thanks the reviewer for the question; the 2 patients who underwent reoperation were affected by HLHS, and reoperation was due to mediastinal revision for bleeding in one ( who had stable hemodynamics), and removal of a clip on the RV-PA conduit in the other. Hemodynamics were anyway stable in both cases.

Comment: During the surgery, what was the minimal hematocrit? Was any relation found between risk for GFAP and hemodilution?

Reply: Hematocrit median value, measured during CEC, was 28% with a minimum value of 25% and a maximum one of 30%. No correlation exists between GFAP maximum value and Hematocrit level measured during CEC (rho=0.0760, p= 0.7718).

Comment: During the surgery, were there any electric brain activity (EEG waves) monitoring performed like CSA?

Reply: Unfortunately, we do not have data on electric brain activity during surgery.

Round 2

Reviewer 2 Report

Dear Authors,

thank you for your answers. I find them convincing.

Although the study populacjom is limited I congratulate you with the study.

Kind regards

TU